# Durability and Mechanical Characteristics of Blast-Furnace Slag Based Activated Carbon-Capturing Concrete with Respect to Cement Content

**Seungwon Kim** [1,2,*] **and Cheolwoo Park** [1,2,*]

1   KIIT (Kangwon Institute of Inclusive Technology), Kangwon National University, 1 Gangwondaegil, Chuncheon 24341, Korea
2   Department of Civil Engineering, Kangwon National University, 346 Jungang-ro, Samcheok 25913, Korea
*   Correspondence: inncoms@kangwon.ac.kr (S.K.); tigerpark@kangwon.ac.kr (C.P.); Tel.: +82-33-570-6518 (S.K.); +82-33-570-6515 (C.P.)

**Abstract:** The recent abnormal temperature phenomena such as the rise of global mean temperature and sea level due to global climate change are clear threats that can no longer be overlooked to the human beings who have pursued indiscriminate development and rapid growth. Climate change has emerged as a serious risk that threatens the survival of the entire human race from the environmental and ecological aspects, despite international efforts for several decades. The $CO_2$ concentration in the atmosphere has increased by approximately 39% since the industrial revolution. Even if carbon emissions are stopped right now, it is expected to take at least 50–200 years to return to the $CO_2$ level before the industrial revolution. Therefore, we conducted an experimental study to develop a carbon-capturing concrete that has active as well as passive carbon reduction functions using blast-furnace slag, an industrial byproduct, instead of cement. For active carbon reduction, we used calcium hydroxide and sodium silicate as carbon capture activators, and conducted tests on mechanical properties and durability characteristics.

**Keywords:** climate change; carbon emission; carbon-capturing concrete; carbon capture activator; carbon reduction

## 1. Introduction

Climate change has emerged as a serious risk factor that threatens the survival of the entire human race from the environmental and ecological aspects, despite exerting the international efforts for several decades [1]. The $CO_2$ concentration in the atmosphere has increased by approximately 39% since the industrial revolution. Even if carbon emissions are stopped right now, it is expected to take at least 50–200 years to return to the $CO_2$ level before the industrial revolution [1]. To cope with this problem, global greenhouse gas (GHG) reduction measures have been made, such as the Kyoto Protocol to the United Nations Framework Convention on Climate Change to tackle global warming [2]. According to the fourth report of the Intergovernmental Panel on Climate Change (IPCC), the global mean temperature has risen by 0.74 °C for the last 100 years and is expected to rise by up to 6.4 °C by the end of the 21st century [1,3].

As such, climate change is a critical global issue, and as one of the world's top 10 countries in GHG emissions, South Korea has set up "carbon reduction technology development" as a new pillar of national development vision, developed related technologies and established laws since 2008 [4]. In addition, South Korea showed the highest annual increasing rate of 6.2% in the transportation sector of GHG emissions from 1990 to 2000 among the Organization for Economic Co-operation and Development (OECD) member countries. To take part in the development of carbon reduction technologies in the

road sector, which accounts for 16% of the total GHG emissions, the need for the development of design and development technologies to absorb and reduce $CO_2$ to minimize $CO_2$ emissions is emerging [4]. Recently, the secondary environmental pollution problem is appearing due to the increase of industrial byproducts such as mining byproducts and coal incineration ashes [4]. For efficient processing of industrial byproducts, some of the industrial byproducts are mixed with existing cement or asphalt materials and these mixtures are used as construction materials [4]. Comprehensive Assessment System for Building Environmental Efficiency (CASBEE) in Japan has evaluated the GHG emissions of building materials using the $CO_2$ emission database of major materials since 2010 [5–7]. Building performance is now a major concern of professionals in the building industry and environmental building performance assessment has emerged as one of the major issues in sustainable construction [8–11]. Efforts to reduce $CO_2$ emission are being made through the use of procurement systems for low $CO_2$ emission materials and the application of high-strength concrete at construction sites. As a part of such research, there is a demand for the establishment of a quantitative assessment method of $CO_2$ emission from concrete production to site procurement and a proposal of a $CO_2$ reduction plan at construction sites [12]. However, the utilization of materials for $CO_2$ capture is still insufficient. The development of concrete applying $CO_2$ reduction technology is considered to be a leading technology for the society as a whole, as well as for the construction sector. The MIT Technology Review selected "Green Concrete", which is a $CO_2$ reducing concrete, one of top 10 new technologies in 2010 [4,13].

Therefore, in this study, we aim to develop carbon-capturing concrete using industrial byproducts that can satisfy not only passive carbon reduction function but also active carbon reduction function through the utilization of industrial byproducts. To develop a carbon-capturing concrete using blast-furnace slag powder, which is one of the representative industrial byproducts, we conducted an experimental study on the mechanical and durability characteristics of carbon-capturing concrete composed of blast-furnace slag using carbon capture activator.

## 2. Research Trends of Carbon-Capturing Concrete

The generation amount of blast-furnace slags, a representative industrial byproduct, has been continuously increasing with the accelerated development of the steel industry. If blast-furnace slag powder is used as a raw material of concrete by mixing with cement, approximately 40 million tons of cement per year can be reduced as well as $CO_2$ emissions [14,15].

To reduce $CO_2$ emissions, the main cause of GHGs, the construction industry is striving to develop environment-friendly concretes. Consequently, cementless binders that use industrial byproducts such as blast-furnace slag, flay ash, and metakaolin instead of ordinary Portland cement (OPC) are being actively researched in and outside the country. Technologies related to cementless binders include geopolymer technology using alkaline reaction with clay minerals and alkali-activated binder technology that develops hardened body by stimulating the latent hydraulic activity of slag byproducts [14,15].

The economic efficiency of 50 MPa or lower alkali-activated concrete is moderately higher than that of the equivalent OPC, but the cost of alkali-activated concrete for high-strength concrete is known to be lower by approximately 10–40% compared to that of the OPC concrete with replaced silica fume [14,15].

Furthermore, the alkali-activated concrete can have advantage over the OPC concrete in terms of economic aspect because it uses industrial byproducts, which do not require a firing process for cement production. Moreover, using an alkali stimulant that is easy to handle can further improve economic efficiency. Therefore, the development of a carbon-capturing concrete through the research of geopolymer concrete and alkali-activated concrete of a non-plastic inorganic binder is expected to effectively reduce $CO_2$ emissions [14,15].

## 3. Experiment Overview

### 3.1. Used Materials

This study used type 1 OPC and Table 1 outlines the physical properties and chemical compositions of this cement. Furthermore, blast-furnace slag powder, which is a representative industrial byproduct, was used to reduce the content of OPC. Table 2 outlines the physical properties and chemical composition of the used blast-furnace slag powder.

**Table 1.** Physical properties and chemical composition of ordinary Portland cement.

| Physical Properties | | | | | |
|---|---|---|---|---|---|
| Specific Gravity | Fineness (cm$^2$/g) | Stability (%) | Setting Time (min) | | Ig-Loss (%) |
| | | | Initial | Final | |
| 3.15 | 3400 | 0.10 | 230 | 410 | 2.58 |
| Chemical Composition | | | | | |
| SiO$_2$ (%) | CaO (%) | MgO (%) | SO$_3$ (%) | Al$_2$O$_3$ (%) | |
| 21.95 | 60.12 | 3.32 | 2.11 | 6.59 | |

**Table 2.** Physical properties and chemical composition of blast-furnace slag.

| Physical Properties | | | |
|---|---|---|---|
| Specific Gravity | Fineness (cm$^2$/g) | Flow Ratio (%) | Ig-loss (%) |
| 2.90 | 4314 | 104 | 0.22 |
| Chemical Composition | | | |
| MgO (%) | SO$_3$ (%) | Chloride Ion (%) | Basicity |
| 3.82 | 1.58 | 0.003 | 1.76 |

The maximum size of 25 mm was used for the coarse aggregates in the carbon-capturing concrete. Tables 3 and 4 outline the physical properties of the used coarse and fine aggregates, respectively.

**Table 3.** Physical properties of the used coarse aggregates.

| Gmax (mm) | Specific Gravity | Water Absorption (%) | Fineness Modulus |
|---|---|---|---|
| 25 | 2.76 | 0.45 | 6.72 |

**Table 4.** Physical properties of the used fine aggregates.

| Specific Gravity | Water Absorption (%) | Fineness Modulus |
|---|---|---|
| 2.52 | 1.45 | 2.62 |

To improve the carbon absorption performance of the carbon-capturing concrete composed of blast-furnace slag, calcium hydroxide ($Ca(OH)_2$) and powder-type sodium silicate ($Na_2SiO_3$) with 95% or higher purity were used as carbon capture activators.

For chemical admixture, a polycarboxylate high range water reducer (HRWR) that has excellent dispersion effect and can achieve fluidity even at a low water-cement ratio was used. Table 5 outlines the properties of the used polycarboxylate HRWR.

**Table 5.** Properties of polycarboxylate high range water reducer (HRWR).

| Principal Component | Specific Gravity | pH | Alkali Content (%) | Chloride Content (%) |
|---|---|---|---|---|
| Polycarboxylate | 1.05 ± 0.05 | 5.0 ± 1.5 | less than 0.01 | less than 0.01 |

### 3.2. Mixing

A literature review for the development of the carbon-capturing concrete composed of blast-furnace slag revealed that replacing part of the blast-furnace slag powder with cement had positive effects in terms of the durability and carbon absorption performance [3,4,15]. Furthermore, the difference in strength of the carbon-capturing concretes by curing temperature was insignificant. Hence, the experimental specimens were cured in a constant temperature and humidity room (23 °C, relative humidity 50%).

Table 6 shows the mix table used in this study. For the carbon-capturing concrete satisfying the mechanical and durability characteristics, the cement replacement ratio relative to the weight of the blast-furnace slag was increased in 10% intervals from 10% to 40%. The water–binder ratio was fixed at 0.325, the fine aggregate ratio at 0.45, and the replacement ratio of carbon capture activator at 40% (calcium hydroxide 20% and sodium silicate 20%) for this experiment.

**Table 6.** Mix variables of carbon-capturing concrete.

| Variables | W/B (%) | S/a (%) | Unit Weight (kg/m$^3$) | | | | | | | Admixture (HRWR) |
| | | | W | BFS | Cement | Activator Ca(OH)$_2$ | Na$_2$SiO$_3$ | F.A. | C.A. | |
| S90Ca20Na20 | | | | 495 | 55 | | | 472.44 | 630.12 | |
| S80Ca20Na20 | 0.325 | 0.45 | 250.3 | 440 | 110 | 110 | 110 | 474.19 | 632.40 | 10.01 |
| S70Ca20Na20 | | | | 385 | 165 | | | 475.85 | 634.68 | |
| S60Ca20Na20 | | | | 330 | 220 | | | 477.54 | 636.90 | |

W/B: Water/Binder (BFS+C+Activator), W: Water, BFS: Blast-Furnace Slag, C: OPC, Ca(OH)$_2$: Calcium Hydroxide, Na$_2$SiO$_3$: Sodium Silicate. F.A.: Fine Aggregate, C.A.: Coarse Aggregate. Admixture: Polycarboxylate High Range Water Reducer. S00: BFS/C, Ca: Ca(OH)$_2$/BFS+C, Na: Na$_2$SiO$_3$/BFS+C.

### 3.3. Experiment Method

An experiment was conducted to analyze the properties of carbon-capturing concrete using blast-furnace slag powder, which is a representative industrial byproduct, and calcium hydroxide and sodium silicate as carbon capture activators before and after curing.

#### 3.3.1. Properties of Concrete before Curing

Slump and air content tests were performed to examine the properties of concrete before curing.

#### 3.3.2. Properties of Concrete after Curing

Exposure Conditions

A literature review revealed that in the high concentration exposure condition of 20% (200,000 ppm) $CO_2$, the carbon capture depth of alkali-activated concrete was approximately 5 mm or lower in 1 month and approximately 20 mm or lower in 12 months [16]. As shown in this literature review, the change of carbon capture depth was insignificant even though the concrete was exposed to a high concentration of $CO_2$ for a long time. In the present study, a $CO_2$ concentration of 10% (100,000 ppm) was set as the high concentration $CO_2$ exposure condition for the accelerated carbonation tester. The relative humidity and temperature inside the accelerated carbonation tester during exposure were 50% and 23 ± 2 °C, respectively.

Furthermore, in the case of the high-purity air exposure conditions for comparison with high concentration $CO_2$ exposure conditions, an exposure experiment was performed in a constant temperature and humidity room at the relative humidity and temperature of 50% and 23 ± 2 °C using a plastic chamber that can be fully sealed by applying a urethane foam solution. The used high-purity air was composed of only nitrogen and oxygen and contained less than 0.01% of other

components ($N_2+O_2 > 99.99\%$), which was considered as a control for comparison with the high concentration $CO_2$ exposure conditions.

Compressive Strength

Cylindrical specimens of $\phi 100$ mm $\times$ 200 mm were fabricated for the compressive strength experiment. To measure the compressive strength for each variable and each exposure condition, the compressive strength characteristics of carbon-capturing concrete were analyzed using the specimens at 1, 7, 28, 56, and 90 days of exposure.

Flexural Strength

Prismatic specimens of 100 mm $\times$ 100 mm $\times$ 400 mm were fabricated for the flexural strength experiment. The flexural strength characteristics of the specimens were analyzed at 7 and 28 days of exposure.

Carbon Capture Depth

To analyze the carbon capture depth of carbon-capturing concrete, the carbon capture depth was measured with respect to the progress of exposure period for each exposure conditions and variables at 1, 7, 28, 56, and 90 days.

Freeze–Thaw Resistance

A freeze–thaw resistance experiment was conducted as an indoor experiment to measure the freeze–thaw resistance of concrete specimens in rapid iterative cycles. The same prismatic specimens of 100 mm $\times$ 100 mm $\times$ 400 mm as those used in the flexural strength experiment were fabricated and cured for 14 days. Then their freeze–thaw resistance was evaluated in accordance with ASTM C 666, Standard Test Method for Resistance of Concrete to Rapid Freezing and Thawing [17].

Chloride Ion Penetration Resistance

For the specimens, the penetration resistance of chloride ion was evaluated in accordance with ASTM C 1202 'Standard Test Method for Electrical Indication of Concrete's Ability to Resist Chloride Ion Penetration' [18], which is a standard test method that determines the electric conductivity to measure the resistance of concrete to chloride ion penetration in a short time. The same cylindrical specimens of $\phi 100$ mm $\times$ 200 mm as the compressive strength specimens were fabricated and the experiment was performed by cutting the specimens to a thickness of $50\pm3$ mm after curing for 28 days. To prevent the water evaporation from the inside of the specimens, the edges of the specimens were coated with a concrete protective coating, and the resistance voltage was measured every 30 min for 6 h. In addition, the passed charge was calculated using Equation (1). Table 7 outlines the evaluation criteria for chloride ion penetration resistance presented in ASTM C 1202 [18].

$$Q = 900 \times \left(I_0 + 2I_{30} + 2I_{60} + \cdots + 2I_{300} + 2I_{330} + 2I_{360}\right) \tag{1}$$

**Table 7.** Evaluation criteria for chloride ion penetration resistance [18].

| Passed Charge (C) | Chloride Ion Penetrability |
|:---:|:---:|
| >4000 | High |
| 2000–4000 | Moderate |
| 1000–2000 | Low |
| 100–1000 | Very Low |
| <100 | Negligible |

Here, Q is the passed charge (C), $I_0$ is the current (A) immediately after applying the voltage, and $I_t$ is the current (A) at t min after applying the voltage [18].

Image Analysis

The air-void structure of the carbon-capturing concrete for each variable with respect to the cement replacement ratio was analyzed using the linear traverse method, which is the A experiment method in ASTM C 457, 'Standard Test Method for Microscopical Determination of Parameters of the Air-Void System in Hardened Concrete' [19]. Furthermore, the air-void structure analysis of the concrete after curing is a method that can be used for indirect evaluation of freeze–thaw resistance.

The image analysis experiment method is often used for rapid measurement and error reduction in comparison with the conventional analysis method by naked eye. This experiment can obtain the total air volume, specific surface area, spacing factor, air volume per air void size, and the number of voids by measuring the sizes and positions of air voids on the images captured by the analysis system. Figure 1 shows the image analysis process.

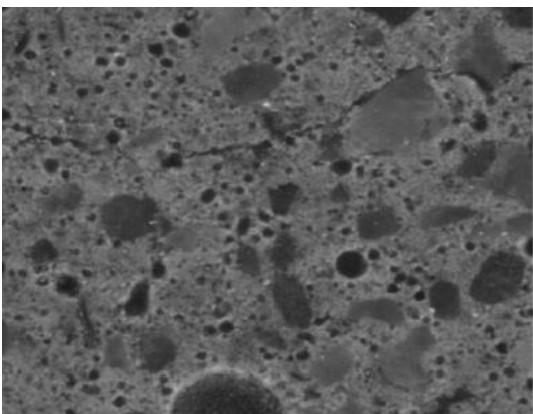 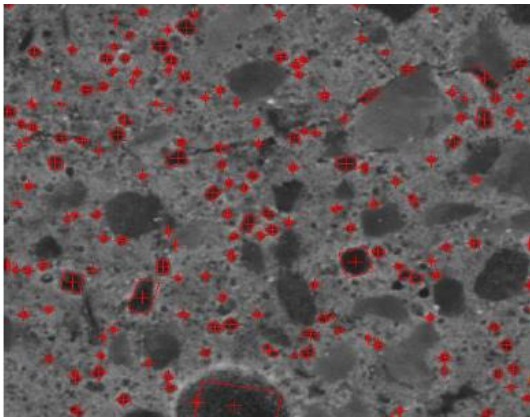

**Figure 1.** Image analysis process.

## 4. Experiment Results and Analysis

### 4.1. Properties of Concrete before Curing

In the slump experiment results of each variable with respect to the cement replacement ratio of carbon-capturing concrete composed of blast-furnace slag, the S90Ca20Na20 variable showed the highest slump value of approximately 45 mm, where the cement replacement ratio was 10%. In the variables where the cement replacement ratio increased in 10% unites from 20% to 40%, the slump values were similar although they increased with the rising cement replacement ratio. Thus, the slump differences by the variable of the carbon-capturing concrete with respect to the cement replacement ratio were insignificant in general.

The S60Ca20Na20 variable, which is the highest cement replacement ratio of the carbon-capturing concrete composed of blast-furnace slag relative to the weight of blast-furnace slag, showed the highest air volume of 3.6%. The changes in the air volume of the carbon-capturing concrete with rising cement replacement ratio were around 3.0% in general. As with the slump experiment results, the differences in air volume by the cement replacement ratio were insignificant. The reason for this phenomenon is that the polycarboxylate HRWR used as an admixture in this study did not have a positive effect on the acquisition of air volume when compared with admixtures exhibiting air entraining performance like air-entraining admixtures. Figure 2 shows the slump and air content test results for each variable.

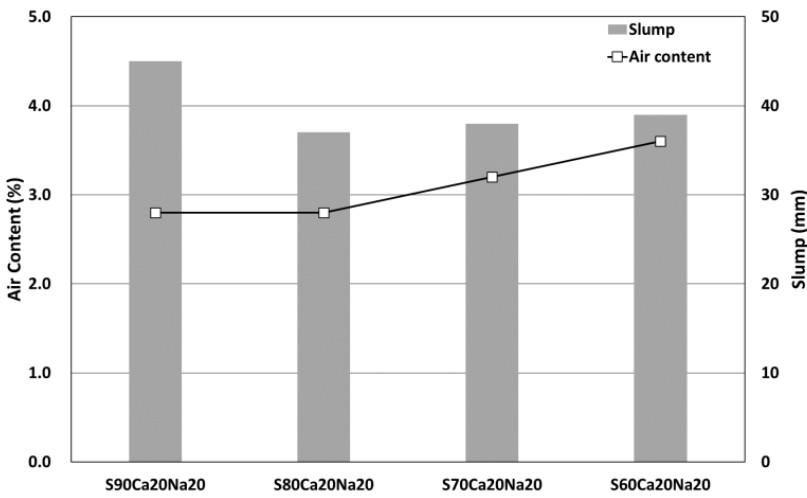

**Figure 2.** Slump and air content test results.

### 4.2. Properties of Hardened Concrete

#### 4.2.1. Compressive Strength

Figure 3 shows the results of compressive strength experiment with respect to the cement replacement ratio of the carbon-capturing concrete composed of blast-furnace slag exposed to high concentration $CO_2$. The S90Ca20Na20 variable with the lowest cement replacement ratio relative to the weight of blast-furnace slag showed the lowest compressive strength of the range from 1 to 90 days of exposure among all the variables. This seems to be due to the differences in the strength expression by the low cement replacement ratio.

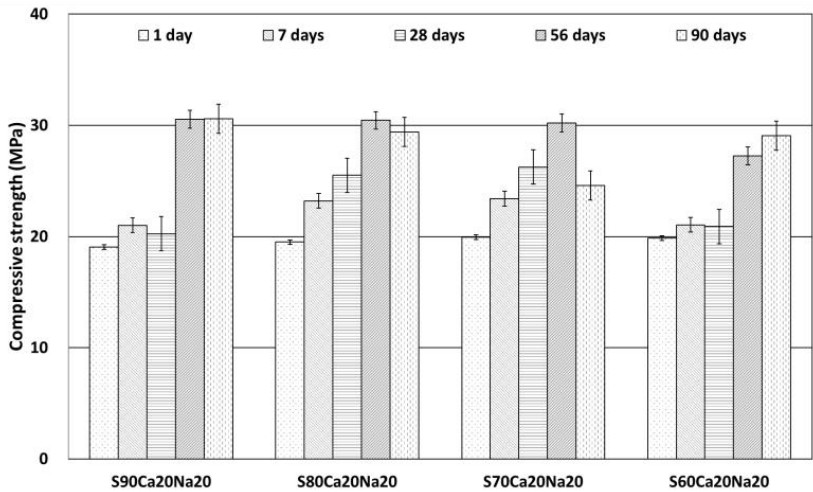

**Figure 3.** Compressive strength experiment results of high concentration $CO_2$ exposure conditions.

The variables with 20%, 30%, and 40% cement replacement ratios showed compressive strengths of around 20 MPa until 90 days of exposure. The increase in the strength with respect to the progress of the exposure period was generally insignificant. The S60Ca20Na20 variable with 40% cement replacement ratio showed 25 MPa or higher compressive strengths at 90 days of exposure. This is due to the differences in strength depending on the higher cement replacement ratio compared to other variables.

Figure 4 shows the results of compressive strength experiment with respect to the cement replacement ratio of the carbon-capturing concrete composed of blast-furnace slag exposed to

the high-purity air. In the experiment results at 1 day of exposure, every variable showed a compressive strength of 20 MPa. Subsequently, until 28 days of exposure, the compressive strength was approximately 25 MPa in the S80Ca20Na20 and S70Ca20Na20 variables with cement replacement ratios of 20% and 30%, respectively. However, the S60Ca20Na20 and S90Ca20Na20 variables showed an insignificant increase in strength.

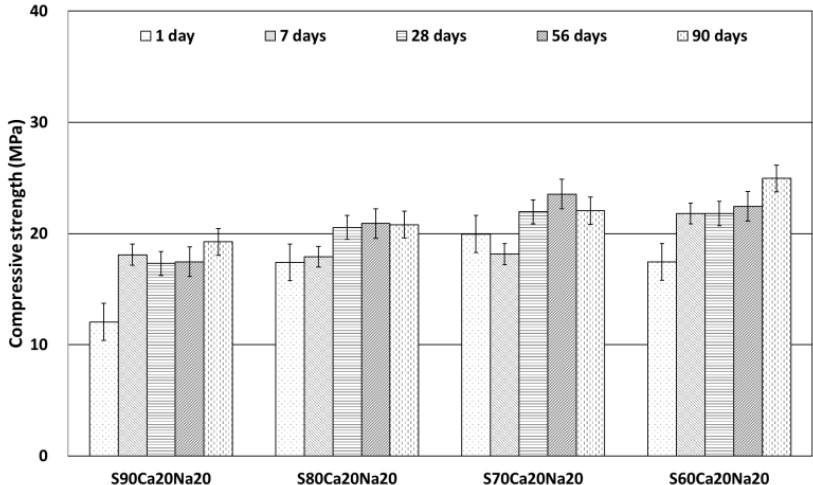

**Figure 4.** Compressive strength experiment results of high-purity air exposure conditions.

At 56 days of exposure, the strength improved by approximately 15% compared to 28 days of exposure. At 90 days of exposure, the strength increase over the exposure period compared to 56 days of exposure was generally insignificant. The S70Ca20Na20 variable with 30% cement replacement ratio relative to the weight of blast-furnace slag showed a decrease of strength at 90 days of exposure. This seems to be due to poor compaction, which is a factor of the specimen fabrication process.

The increase of the compressive strength of the carbon-capturing concrete with respect to the cement replacement ratio and exposure conditions over the exposure period was generally insignificant. The variable with the high-purity air condition showed a greater increase of the compressive strength than the variable of exposure in high concentration $CO_2$ condition. This is considered to be due to the difference in the relative strength expression speed with the variable of high-purity air exposure generated by contraction due to the surface carbon absorption of the specimen exposed to high concentration $CO_2$.

### 4.2.2. Flexural Strength

Figure 5 shows the results of flexural strength experiment for each exposure condition and variable of carbon-capturing concrete composed of blast-furnace slag at 7 days of age. In the case of 7 days of exposure, the flexural strength in the high concentration $CO_2$ exposure condition was approximately 3.0 MPa. The difference in flexural strength with the rising cement replacement ratio was insignificant. Furthermore, the variable of high-purity air exposure condition showed a flexural strength of approximately 4.0 MPa, which is higher than that of the $CO_2$ exposure condition.

Figure 6 shows the flexural strength experiment results for each exposure condition and variable of carbon-capturing concrete composed of blast-furnace slag at 28 days of age. In the case of 28 days of exposure, the flexural strength in the high concentration $CO_2$ exposure condition was approximately 3.0 MPa. The S60Ca20Na20 variable of 40% cement replacement ratio showed the highest flexural strength of 3.2 MPa. Furthermore, the strength increase compared to the result of 7 days of exposure for the high concentration $CO_2$ condition was insignificant.

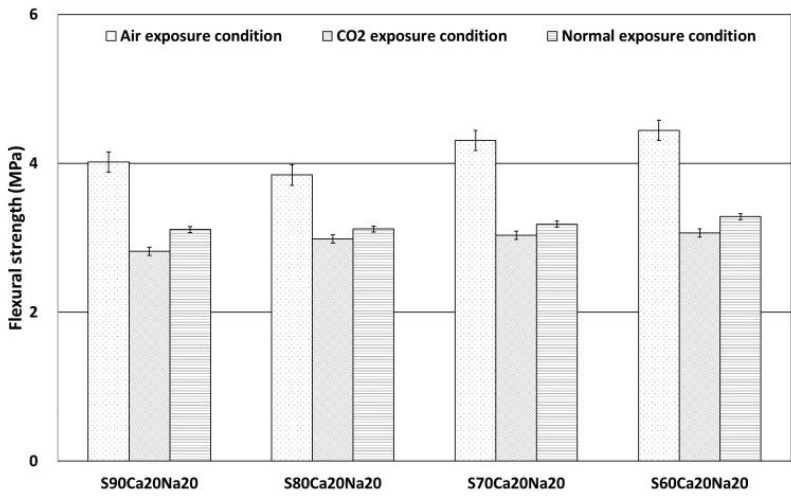

**Figure 5.** Flexural strength experiment results (7 days of age).

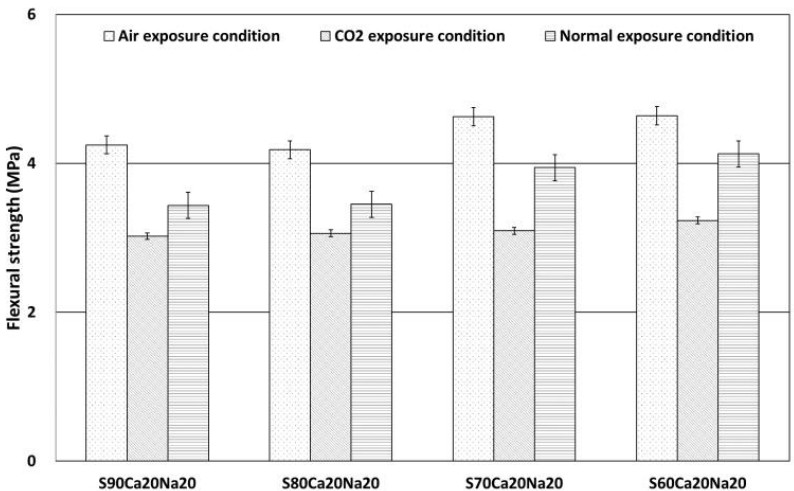

**Figure 6.** Flexural strength experiment results (28 days of age).

In the high-purity air exposure condition, the flexural strength was measured above approximately 4.0 MPa in general. The variable of a higher cement replacement ratio also showed a higher flexural strength. When compared to 7 days of exposure, the strength improved by approximately 5%. The increase of flexural strength for each variable with respect to the exposure condition and cement replacement ratio of the carbon-capturing concrete over the exposure period was moderately insignificant. Furthermore, compared to the high concentration $CO_2$ exposure condition, the increase of the variable for measurement in high-purity air exposure condition was higher compared to the high concentration $CO_2$ exposure condition. This seems to be due to the difference in the strength expression speed generated because the effect of carbon absorption is smaller than the high-concentration $CO_2$ exposure condition. Furthermore, the difference in flexural strength due to the increasing cement replacement ratio of the carbon-capturing concrete was moderately insignificant.

### 4.2.3. Carbon Capture Depth

Figure 7 shows the carbon capture depth measurement result for each exposure period of the exposed variable that was expressed in high concentration $CO_2$. No change in depth according to carbon capture did not appear at one day, but at 7 days, a carbon capture depth of more than approximately 5 mm was measured. At 28 days of exposure, a carbon capture depth of approximately

10 mm appeared. Until 90 days of exposure, approximately 20 mm carbon capture depth appeared in the S60Ca20Na20 variable with the highest 40% cement replacement ratio.

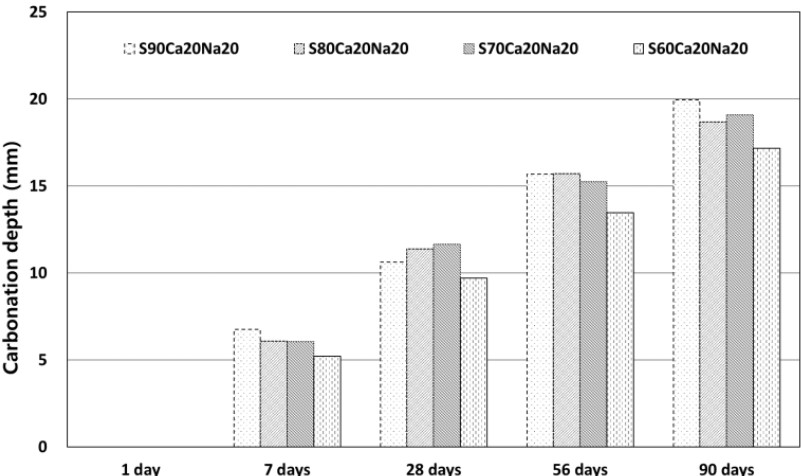

**Figure 7.** Carbon capture depth measurement result (exposure to high concentration $CO_2$).

Furthermore, the replacement ratio of blast-furnace slag powder increased with the exposure period, and the carbon capture depth increased with declining cement replacement ratio. This is considered to be due to the increased carbonation resistance through the generation of calcium hydroxide by calcium silicate by the rising cement replacement ratio.

The carbon capture depth measurement result of the high-purity air exposure conditions variable showed that no change in the carbon capture depth occurred in every variable from 1 to 90 days of exposure. This is considered to be due to no interference from external $CO_2$ resulting from the injection of air consisting of $N_2$ and $O_2$ only in the sealed chamber in the case of high-purity air exposure condition.

The carbon capture depth did not change at 1 day of exposure in the case of the high concentration $CO_2$ exposure condition. However, the maximum carbon capture depth was approximately 20 mm from the 7 to 90 days of exposure during which carbon capture from the surface occurred continuously. In the case of the high-purity air exposure condition, the carbon capture did not change due to the progress of exposure until 90 days of exposure. This result is considered to be due to the carbonation resistance resulting from not receiving direct effect of $CO_2$ exposure compared to the high-concentration $CO_2$ exposure condition.

### 4.2.4. Freeze–Thaw Resistance

Figure 8 shows the results of freeze–thaw resistance experiment with respect to the cement replacement ratio of the carbon-capturing concrete. The relative dynamic modulus of elasticity increased to much higher than 80% for 300 freeze–thaw cycles, but no difference in the freeze–thaw resistance was caused by the change of the cement replacement ratio.

This is considered to be due to the positive freeze–thaw resistance resulting from the satisfaction of the spacing factor (200 μm or less) and specific surface area (25 mm$^2$/mm$^3$ or higher) conditions as in the result of the image analysis (air-void property analysis).

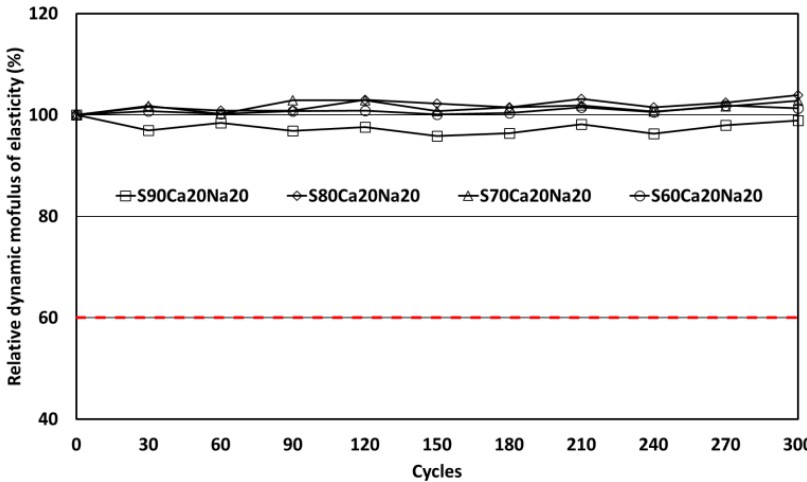

**Figure 8.** Freeze–thaw resistance experiment results.

### 4.2.5. Chloride Ion Penetration Resistance

Figure 9 shows the experiment results for penetration resistance of chloride ion in the carbon-capturing concrete with respect to the cement replacement ratio. The S90Ca20Na20 variable with the lowest cement replacement ratio of 10% relative to the weight of blast-furnace slag showed the lowest total passed charge of 1264 (coulombs). Subsequently, the total passed charge tended to increase with a declining cement replacement ratio. In the case of the S60Ca20Na20 variable with the highest cement replacement ratio of 40%, the total passed charge was measured at 1544 (coulombs), which is higher by approximately 22% than that of the S90Ca20Na20 variable.

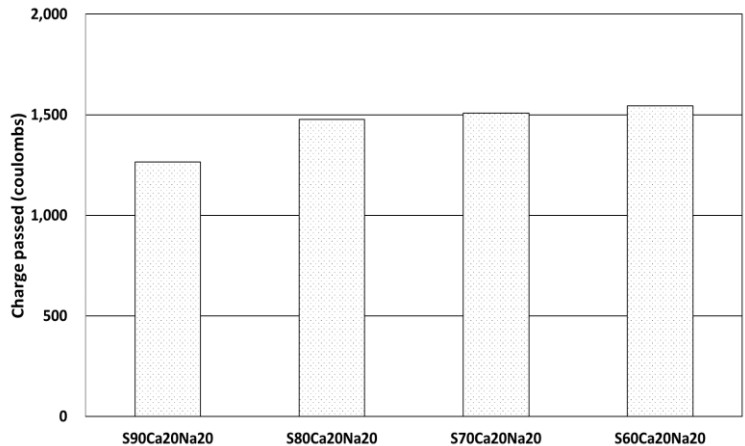

**Figure 9.** Chloride ion penetration resistance experiment results.

The penetration resistance of chloride ion in the carbon-capturing concrete was evaluated as 'low' as the passed charge ranged from 1000 to 2000. It was analyzed that the increased water-tightness of the voids in the concrete due to the mixing of blast-furnace slag powder increased the chloride ion penetration resistance. Furthermore, the differences in chloride ion penetration due to the increase of cement replacement ratio resulting from the use of blast-furnace slag above a certain level were generally insignificant.

### 4.2.6. Image Analysis

Figure 10 shows the spacing factor and specific surface area measurement results obtained through the image analysis of carbon-capturing concrete with respect to the cement replacement ratio.

The spacing factor was generally lower than 185 μm, and the change of the spacing factor with respect to the cement replacement ratio was moderately insignificant. Furthermore, every variable showed a specific surface area above 30 mm$^2$/mm$^3$, and similar levels of specific surface areas were measured in general.

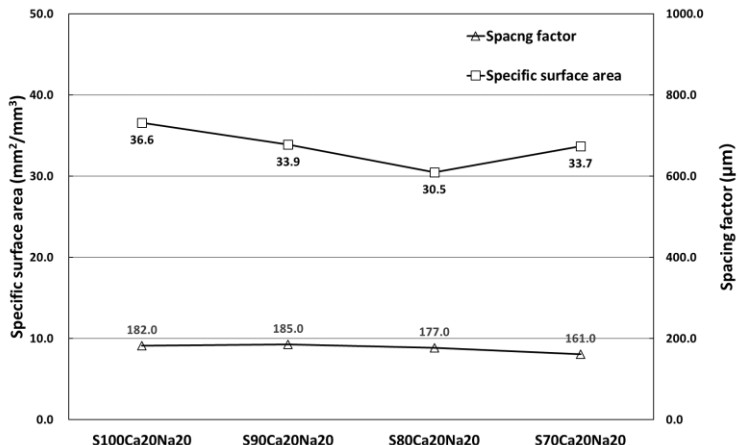

**Figure 10.** Image analysis experiment results.

The spacing factor and specific surface area of the voids of the concrete after curing are factors for forming good air-void structure of concrete. Sidney Mindess [20] reported that the spacing factor of concrete after curing must be 200 μm (0.2 mm) or lower and the specific surface area 25 mm$^2$/mm$^3$ or higher to secure sufficient freeze–thaw resistance [20]. Therefore, the freeze–thaw resistance of the carbon-capturing concrete with respect to the cement replacement ratio is considered to be excellent based on the measured spacing factor and specific surface area.

## 5. Conclusions

An experimental study was conducted to develop a carbon-capturing concrete composed of blast-furnace slag that can perform passive as well as active carbon reductions by replacing cement with industrial byproducts. To examine the applicability of this carbon-capturing concrete, we conducted experiments about the mechanical and durability characteristics with respect to the cement replacement ratio. The conclusions of this study are as follows.

(1) The compressive strength experiment results with respect to the exposure conditions and cement replacement ratio showed that the compressive strength of the specimen exposed to high concentration $CO_2$ decreased very little over the exposure period in general. In the high-purity air exposure condition, the compressive strength increased greater than that of the high concentration $CO_2$ exposure condition. However, the increase of compressive strength with respect to the cement replacement ratio was insignificant.

(2) The flexural strength experiment results showed that the increase of the flexural strength with respect to the cement replacement ratio was insignificant as in the compressive strength experiment results. The high-purity air exposure condition showed a higher flexural strength than that of the high concentration $CO_2$ exposure condition. This is considered to be due to the difference in the strength expression speed caused by the relative lack of the carbon absorption effect.

(3) The carbon capture depth experiment result of the high concentration $CO_2$ exposure condition showed that the carbon capture depth changed very little with the rising cement replacement ratio, and almost no carbon capture occurred in the high-purity air exposure condition.

(4) The freeze–thaw resistance experiment results showed that the air volume of the concrete before curing was approximately 3%, but the relative dynamic modulus of elasticity increased to much

higher than 80% until 300 freeze–thaw cycles. No change was caused by the cement replacement ratio. The image analysis for the air-void structure showed that every variable satisfied the condition of 200 μm (0.2 mm) or lower spacing factor and 25 $mm^2/mm^3$ or higher specific surface area, which are the criteria for securing freeze–thaw resistance. Despite somewhat low air volume experiment result, the freeze–thaw resistance was excellent owing to the spacing factor of air bubbles and the entrained air.

(5)    The experiment results of chloride ion penetration resistance showed that the passed charge increased with rising cement replacement ratio, but the difference was not large. Furthermore, every variable showed a 'low' value in the total passed charge range of 1000–2000 coulomb, indicating excellent chloride ion penetration resistance. This is considered to be due to the increased water-tightness of concrete by the mixing of a large quantity of blast-furnace slag despite the cement replacement.

(6)    To summarize the experiment results on the mechanical and durability characteristics of the carbon-capturing concrete composed of blast-furnace slag, the carbon-capturing concrete had mechanical and durability characteristics above the appropriate levels for concrete. Furthermore, this study found that the active as well as passive carbon reduction functions can be achieved through the use of blast-furnace slag, an industrial byproduct, and the development of the carbon-capturing concrete can effectively reduce $CO_2$ emissions.

**Author Contributions:** Conceptualization, S.K. and C.P.; methodology, S.K. and C.P.; writing—original draft preparation, S.K. and C.P.; writing—review and editing, S.K. and C.P. All authors have read and agreed to the published version of the manuscript.

**Funding:** This study was conducted under research project 'Development of High-Performance Concrete Pavement Maintenance Technology to Extend Roadway Life (20POQW-B146693-03)' funded by the Ministry of Land, Infrastructure and Transport (MOLIT) and the Korea Agency for Infrastructure Technology Advancement (KAIA).

**Conflicts of Interest:** The authors declare no conflicts of interest.

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
