# Peer review of "Durability and Mechanical Characteristics of Blast-Furnace Slag Based Activated Carbon-Capturing Concrete with Respect to Cement Content"

_applsci, doi:10.3390/app10062083_

Round 1

Reviewer 1 Report

Well written and scientifically sound paper,  Parts needs to open for improvement are as follows: as title states authors claims concrete in the study is carbon capturing , are they conducted any test on carbon dioxide capturing tests. If so, test results needed to be added to support their arguments. Tables are very difficult to follow, they need to be formatted accordingly and placed in the text properly. Minor grammar checks are recommended.

Author Response

Response to Reviewer 1 Comments

Well written and scientifically sound paper, Parts needs to open for improvement are as follows:

Point 1:

as title states authors claims concrete in the study is carbon capturing, are they conducted any test on carbon dioxide capturing tests. If so, test results needed to be added to support their arguments.

Response 1:

Thank you for your comments.

The results of carbon capture test were mentioned in Section 4.2.3.

Point 2:

Tables are very difficult to follow, they need to be formatted accordingly and placed in the text properly.

Response 2:

Thank you for your comments.

The tables were arranged and placed in the text.

Point 3:

Minor grammar checks are recommended.

Response 3:

Thank you for your comments.

It had been corrected.

Reviewer 2 Report

The paper entitled "Durability and Mechanical Characteristics of Blast Furnace Slag based Activated Carbon-Capturing Concrete with respect to Cement Content" presents an interesting study in relation to the capacity of CO2 capture of concrete. The paper is well structured and the writing is adequate. In addition, the results presented are a good basis for future research based on this.

Line 91 Don't use units with the values, use them in the head file of the table
Line 167 use the symbol of the unit in the International System. C instead coulomb and A instead ampere
Line 168 same comment
Line 169 same comment
Line 203 Use "4.2 Properties of hardened concrete" instead "4.2 Properties of concrete after curing"
Line 321 Use numbering for Image analysis
Line 326 use super index in mm2/mm3

In my opinion the paper can be accepted with minor changes.

Author Response

Response to Reviewer 2 Comments

The paper entitled "Durability and Mechanical Characteristics of Blast Furnace Slag based Activated Carbon-Capturing Concrete with respect to Cement Content" presents an interesting study in relation to the capacity of CO2 capture of concrete. The paper is well structured and the writing is adequate. In addition, the results presented are a good basis for future research based on this.

Point 1:

Line 91 Don't use units with the values, use them in the head file of the table

Response 1:

Thank you for your comments.

It had been corrected.

Point 2:

Line 167 use the symbol of the unit in the International System. C instead coulomb and A instead ampere.

Response 2:

Thank you for your comments.

It had been corrected.

Point 3:

Line 168 same comment

Response 3:

Thank you for your comments.

It had been corrected.

Point 4:

Line 169 same comment

Response 4:

Thank you for your comments.

It had been corrected.

Point 5:

Line 203 Use "4.2 Properties of hardened concrete" instead "4.2 Properties of concrete after curing"

Response 5:

Thank you for your comments.

It had been changed.

Point 6:

Line 321 Use numbering for Image analysis

Response 6:

Thank you for your comments.

It had been corrected.

Point 7:

Line 326 use super index in mm2/mm3

Response 7:

Thank you for your comments.

It had been corrected.

Point 8:

In my opinion the paper can be accepted with minor changes.

Response 8:

Thank you for your comments.

Manuscript had been updated.

Reviewer 3 Report

The article is interesting and include important results.

There are minor language mistake, e.g. it is common use 'flexural strength' not bending.

The author use only 12 position of literaturÄ™, there is a lot of papers which could be cited with this topic.

I think that the author should include more widely literature review.

Author Response

Response to Reviewer 3 Comments

The article is interesting and include important results.

Point 1:

There are minor language mistake, e.g. it is common use 'flexural strength' not bending.

Response 1:

Thank you for your comments.

Bending strength is replaced by flexural strength and other words had been updated.

Point 2:

The author use only 12 position of literaturÄ™, there is a lot of papers which could be cited with this topic.

Response 2:

Thank you for your comments.

Few references had been added.

Point 3:

I think that the author should include more widely literature review.

Response 3:

Thank you for your comments.

It had been updated.

Round 2

Reviewer 3 Report

Good job!

Best wishes.